# Exploring factors associated with dual tobacco smoking among people living with HIV receiving care at HIV outpatient clinics in Hanoi, Vietnam

Thanh Ha-Lan Hoang[1]*, Gloria Guevara Alvarez[2], Louise Adermark[3], Nawi Ng[1], Trang Nguyen[4], Nam Nguyen[4], Donna Shelley[2]

**1** School of Public Health and Community Medicine, Institute of Medicine, Sahlgrenska Academy, University of Gothenburg, Gothenburg, Sweden, **2** New York University School of Global Public Health, New York, New York, United States of America, **3** Department of Pharmacology, Institute of Neuroscience and Physiology, Sahlgrenska Academy, University of Gothenburg, Gothenburg, Sweden, **4** Institute of Social and Medical Studies, Hanoi, Vietnam

* thanh.hoang@gu.se

## Abstract

### Background

People living with HIV (PLWH) who smoke face significant health risks from tobacco use, which dual use of cigarettes and waterpipes may exacerbate due to increased nicotine exposure and dependency. This study examined the sociodemographic, behavioral, psychosocial and health-related factors associated with dual smoking among PLWH receiving care at HIV clinics in Hanoi, Vietnam.

### Methods

This cross-sectional study used baseline survey data from the VQUIT study, an RCT of smoking cessation interventions involving 662 PLWH from 13 outpatient clinics in Vietnam. Eligible participants were current tobacco smokers; dual smokers were those using both cigarettes and waterpipes. Multivariable logistic regression was used to identify factors associated with cigarette-only and dual use among PLWH.

### Results

Over half of the study participants were dual smokers. Dual smoking was associated with a lower annual income (aOR: 1.91, 95%CI: 1.08–3.38) and higher nicotine dependence (aOR: 2.31, 95%CI: 1.62–3.31). They were less likely to report a prior quit attempt (aOR: 0.47, 95% CI: 0.33–0.68), more likely to perceive cigarettes as more harmful (aOR 1.44, 95% CI: 1.01–2.09), and more likely to report recent illicit drug use (aOR: 2.32, 95% CI: 1.26–4.24) and longer antiretroviral treatment duration (aOR: 1.06, 95% CI: 1.03–1.10). Social support for quitting lowered the odds of dual smoking (aOR: 0.62, 95% CI: 0.39–0.97).

**Data availability statement:** The dataset underlying this study contains identifiable and highly sensitive health information (including HIV status). In accordance with determinations by the Institutional Review Boards at New York University and the Institute of Social and Medical Studies (ISMS), Hanoi, and applicable ethics approvals, the data cannot be placed in a public repository. Qualified researchers may request access to a de-identified/minimal dataset for reproducibility purposes by contacting: New York University – Institutional Review Board (IRB) Scott Fisher, Director, Human Research Compliance Email: srf291@nyu.edu Web: NYU IRB Office Requests must include a brief proposal outlining the intended use and analytic plan. Requests are reviewed independently by the NYU IRB for compliance with participant confidentiality and data-protection policies. If approved, requestors (and their institution) must sign a data-use agreement; additional local IRB/EC approvals may also be required. No author will adjudicate access decisions. Location/citation of the dataset. The dataset is held by the New York University IRB and ISMS Ethics Committee and is not publicly archived. Please cite as: Hoang T, et al. Exploring factors associated with dual tobacco smoking among people living with HIV receiving care at HIV outpatient clinics in Hanoi, Vietnam. Institute of Social and Medical Studies, Hanoi & New York University. 2020. Dataset available on approved request from the NYU IRB (contact above).

**Funding:** This study was supported by the National Cancer Institute under Award Number R01CA24481. TH received a doctoral studentship at Sahlgrenska Academy to conduct the research. There was no additional external funding received for this study. The funders had no role in study design, data collection and analysis, decision to publish, or preparation of the manuscript.".

**Competing interests:** The authors have declared that no competing interests exist.

## Conclusions

Dual smoking was prevalent among PLWH receiving HIV care in VQUIT study and was shaped by complex sociodemographic, behavioral, and social factors. These findings emphasize the need for integrated cessation interventions in Vietnam, where dual smoking is prevalent, particularly among PLWH. Socioeconomic status and barriers to quitting, including drug use, nicotine dependence, and tobacco-related misconceptions, should be considered. Future research should examine the long-term health impacts of dual smoking and evaluate cessation strategies for PLWH in similar settings.

## Introduction

Tobacco use remains a significant public health concern worldwide, particularly among vulnerable populations such as people living with HIV (PLWH). Tobacco use is highly prevalent in PLWH and poses additional health risks to them, such as increased susceptibility to cancers, cardiovascular and respiratory diseases [1–4]. Despite significant progress in HIV prevention and treatment, with prevalence decreasing [5], the smoking prevalence among PLWH in Vietnam is alarmingly high at 36%, far exceeding the national average of 22% among adults [6,7]. This highlights the urgent need for targeted interventions to address tobacco use among PLWH.

Cigarette smoking is the most common form of tobacco use and has been widely studied. However, dual- or poly-tobacco use, including waterpipe smoking, presents new challenges due to heightened nicotine dependence, more contextual smoking triggers, and misconceptions about the harmful effects of tobacco use [8–10]. Existing cessation programs often lack strategies tailored to address the complexities of dual tobacco use, which may lead to lower success rates and increased risk of relapse. Global Adult Tobacco Survey 2020 showed that 10% of tobacco smokers in Vietnam smoked both cigarettes and waterpipes, with dual smoking being more prevalent in northern Vietnam at 29% [11,12]. While data on dual use among PLWH in low-resource settings like Vietnam are scarce, studies from the US reported that poly-tobacco use (i.e., the use of cigarettes, cigars, chew, and snuff) is more common among PLWH than in the general population [13–15]. Even though no surveys have reported the prevalence of dual tobacco use in PLWH in Vietnam, over half of PLWH participants in two qualitative studies reported dual smokers of cigarettes and waterpipes, which underscores this concern [16,17].

Waterpipe smoking, the second most common form of tobacco use in Vietnam, contains significantly higher nicotine levels (9%) compared to cigarettes (1%–3%) [7,18]. Waterpipe tobacco smoke also has higher concentrations of carbon monoxide (CO) and polyaromatic hydrocarbons than cigarette smoke [19]. Despite these elevated risks, waterpipe smoking is often mistakenly perceived as less harmful, contributing to its popularity and dual use [20]. This misconception, coupled with the waterpipe smoking as a common social activity, may contribute to its appeal among PLWH, who often face stigma and social isolation [21]. Dual use exacerbates the

already heightened risks of cardiovascular disease and certain cancers in PLWH, presenting significant challenges for smoking cessation efforts [22,23].

Evidence from high-income countries suggests that factors such as sociodemographic characteristics, discrimination, anxiety, and intention to quit influenced poly-tobacco use among PLWH [13–15]. While these findings provided valuable insights, there is a critical gap in understanding dual smoking among PLWH in LMICs. In Vietnam, where tobacco use is widespread among PLWH, exploring the patterns and associated factors of dual use is essential. Hence, this study aims to explore the factors associated with dual tobacco use among PLWH receiving care at HIV clinics and participating in a smoking cessation intervention in Hanoi, Vietnam [24].

## Methods

### Data source and study population

We used baseline survey data from the VQUIT study collected between 30 November 2020 and 27 September 2023. VQUIT is a three-arm randomized controlled trial of smoking cessation intervention for PLWH (n = 672) [25]. Participants were recruited from 13 HIV outpatient clinics in Hanoi. We included patients ≥ 18 years old who were currently smoking cigarettes or both cigarettes and waterpipes, living in Hanoi, and reachable via phone. Written consent forms were obtained for eligible patients. Patients were excluded if they had any contraindications to nicotine replacement therapy use, such as recent myocardial infarction (2 weeks), serious underlying arrhythmias, pregnancy, or breastfeeding, or were unable to demonstrate capacity for consent or already enrolled in another tobacco use treatment program. After completing the survey, participants received an incentive of 2.5 USD. Patients with incomplete data on the study variables were excluded (n = 10). In total, 662 individuals were included in this study.

### Measurements

**Dependent variables.** Tobacco smokers were identified as individuals who reported smoking exclusively cigarettes or both cigarettes and waterpipes every day or someday. The outcome of interest was the types of tobacco product used, categorized into two groups: *Cigarette-only smokers* and *Dual smokers* (those who smoked both cigarettes and waterpipes).

**Independent variables.**

- Sociodemographic variables

Age in years, sex (*Male* or *Female*), marital status (*Single/Divorced/Separated/Widowed* or *Married*), educational level (*No school/Primary/Secondary school, High school (Grade 10–12)*, or *Vocational training/College/University*), occupation (*Unemployed/Homemaker*, *Salaried/paid job*, or *Others*), and living arrangement (*Live alone, Live with spouse/children/grandchildren* or *Live with others*). Household income in the past 12 months used the cut-off of 300 million Vietnam Dong (VND) to categorize high-income households (≥ *300 million VND)* and lower-income households *(< 300 million VND)*

- Health behavioral variables

Illicit drug use was categorised as *Never, Not in the last 3 months* or *In the last 3 months*. Hazardous drinking was derived from the Alcohol Disorders Identification test (AUDIT-C) and adjusted for gender-specific thresholds (*Yes* if AUDIT-C score ≥ 4 for men and ≥3 for women or *No*) [26,27].

- HIV treatment and health-related variables:

Depressive symptoms were assessed using the Centre for Epidemiological Studies Depression Scale (CES-D 8), ranging from 0 to 24 (*Yes* if CES-D 8 score ≥9 or *No)*, and self-rated health status was categorized as *Fair/Poor* or *Excellent/Very good/Good* [28,29]. Time since diagnosis with HIV and duration of antiretroviral (ARV) treatment were measured in years.

Adherence to ARV treatment was defined using a self-reported Visual Analog Scale, where participants indicated the percentage of doses taken over the past four weeks (*Range: 0–100%*). A threshold of ≥95% was used to define optimal adherence, based on prior research showing that this level was associated with viral suppression in ARV patients [30–32].

- Smoking-related variables

Past attempts to quit cigarettes were categorized as *Never tried to quit* or *Ever tried to quit*. Nicotine dependence was based on the Fagerström test of cigarette and waterpipe dependence (*Very low/Low/Medium* or *High/Very High*) [33,34]. Other variables included receipt of advice to quit from a healthcare provider in the last visit (*No* or *Yes*), smoke-free home rules (*Smoking is allowed everywhere at all times* or *Smoking is not allowed anywhere*/allowed *in some places at some times*), and perceived risk of cigarette smoking compared to waterpipe smoking (*More harmful, Less harmful* or *Equally harmful*).

Smoking risk perceptions were measured by four 4-point Likert scale questions suggested by Kaufman et al. with answers ranging from 0–3 points corresponding to "Not at all", "A little likely", "Very likely" and "Extremely likely", respectively [35]. The four questions include: **1** "If you continue to smoke, how likely do you think it is that you will get a disease related to smoking, like cancer, heart disease?", **2** "How likely is it that continuing to smoke will increase your risk of getting an illness related to HIV?", **3** "In your opinion, how likely is it that quitting smoking would reduce your chances of getting a disease related to smoking, like cancer or heart disease?", **4** "How worried are you about getting cancer, heart disease or other smoking-related diseases?". Our analysis used a summary score of the responses from the four questions (*Range: 0–12*). The higher the scores, the higher the risk perceptions.

Intention to quit smoking cigarettes was defined by the question based on the Transtheoretical Model and Stages of Change [36]: "Please indicate which statement best describes what you think about quitting smoking cigarettes?" Those who answered "not planning/intending to quit" were classified as *Not planning/trying to quit.* Intention to quit (i.e., *Planning/trying to quit*) was defined as answering yes to one of the following options: planning to quit in the next 6 months, or the next 30 days, or trying to quit.

Self-efficacy was measured using the Smoking Abstinence Self-efficacy Questionnaire (SASEQ), a short, reliable, and valid questionnaire to assess self-efficacy beliefs regarding smoking abstinence (*Range: 0–24*) [37].

- Social factors

Social pressure to stop smoking (i.e., having friends/family members/coworkers objecting to smoking; *No* or *Yes*), having smokers in their social networks (*Yes* or No), and the number of smokers in the five closest persons (*Less than three* or *Three or more*) were binary variables. Social support was measured using the Multidimensional Scale of Perceived Social Support scale (MSPSS) (*Range: 1–4*) [38]. Internalized smoking stigma was assessed by 4-point Likert scale questions about the level of agreement on three statements based on the Internalized Stigma of Smoking Inventory, including "I am embarrassed or ashamed that I am a smoker", "I am disappointed in myself for being a smoker", and "I feel inferior to others who are not smokers" [39]. Responses to the three items were summed to generate a total score ranging from 1 to 4, with higher values indicating a stronger internalized stigma. Injunctive norms were assessed by smokers' responses to the statements that people important to them who believed that they should not smoke cigarettes and waterpipes (*Disagree* if one Disagreed/Strongly disagreed with the statement or *Agree* if one Agreed/Strongly agreed to the statement).

### Statistical analysis

Descriptive statistics included means and standard deviations for continuous variables and frequencies and percentages for categorical variables. Logistic regression was used to identify factors associated with types of tobacco product use, using cigarette-only use as the reference category. Bivariate analyses were conducted to obtain crude effect sizes of the identified independent variables on dual smoking. The initial multivariable model included variables with p < 0.2 in bivariate

analyses. We then used a backward stepwise approach, removing variable(s) with p > 0.05 and adjusted ORs (aOR) close to 1 to arrive at the final model. Self-efficacy was excluded from the final model due to aORs close to one and non-significant p-values, and time since HIV diagnosis was excluded due to high correlation with the duration of ARV treatment (r = 0.79). AORs with 95% confidence intervals (95% CI) were calculated. The analysis was conducted using STATA SE Version 18.

### Ethical consideration

The study has received ethical approval from the Institute of Social Medical Studies, Hanoi, Vietnam (00007993), New York University (NYU) Institutional Review Board (s19-01783) and Swedish Ethical Review Authority (dnr 2023-06698-01).

## Results

### Descriptive statistics

Table 1 shows the characteristics of our study participants. A total of 662 participants were included in the study, with 332 (50%) categorized as dual smokers. The mean age of the participants was approximately 44 years for both groups. A higher proportion of dual smokers were male, unmarried and reported secondary education or lower, compared to cigarette-only smokers. More dual smokers reported unemployment and earning less than 300 million VND (about 11,800 USD) annually compared to cigarette-only smokers. Less than 10% of both groups were living alone.
Dual smokers had higher rates of former and recent drug use, poorer self-rated health, and longer durations living with HIV and on ARV treatment compared to cigarette-only smokers. Hazardous drinking was relatively similar (57–58%) in both groups. Over half of all participants reported that smoking was allowed in their homes and received provider advice to quit. More dual smokers believed cigarettes were more harmful than waterpipes (42% vs. 29%) and had higher risk perception but lower self-efficacy. Nicotine dependence was higher among dual smokers (58% vs. 33%). Most in both groups planned to quit smoking. Past quit attempts were more common among cigarette-only smokers (58%) than dual smokers (<40%). For social factors, over 95% in both groups agreed with the norms that important people to them believed that they should not smoke, and around 89% reported that family or peers objected to smoking. While nearly all had smokers in their networks, more dual smokers reported ≥3 close contacts who smoked (76% vs. 67%). Cigarette-only smokers reported slightly more social support for quitting and lower perceived stigma.

### Factors associated with types of tobacco product use

**Bivariate analysis.** Fig 1 presents the factors associated with dual smoking among PLWH in Hanoi, Vietnam. Women and individuals having post-high school education or salaried jobs were less likely to be dual smokers. Dual smoking was associated with a household income lower than 300 million VND (OR: 1.96, 95%CI: 1.18–3.24).

Illicit drug use was associated with a higher likelihood of dual smoking. Smokers who self-rated fair or poor health status had a 47% higher likelihood of being dual smokers (OR: 1.47, 95% CI: 1.05–2.06). Duration of HIV diagnosis (OR: 1.07, 95% CI: 1.04–1.09) and of ARV treatment uptake (OR: 1.05, 95% CI: 1.02–1.08) were positively associated with dual smoking.

Smokers who reported smoke-free home rules and viewed cigarette smoking as less harmful than waterpipes were less likely to be dual smokers. Conversely, smokers who considered cigarettes more harmful (OR: 1.59, 95%CI: 1.15–2.20) were more likely to engage in dual smoking. Having high or very high nicotine dependence (OR: 2.89, 95% CI: 2.11–3.96) and a history of cigarette quit attempts (OR: 0.43, 95% CI: 0.32–0.59) was associated with a lower likelihood of dual smoking. Having a larger social network of smokers (≥ 3) was significantly associated with higher odds of dual smoking (OR: 1.53, 95% CI: 1.09–2.15). Individuals scoring higher on the MSPSS were less likely to engage in dual use (OR: 0.66, 95%CI: 0.45–0.98).

**Table 1. Characteristics of study participants (n = 662).**

| | Type of tobacco product use | |
|---|---|---|
| | **Cigarette-only smokers** | **Dual smokers** |
| | 330 (49.8%) | 332 (50.2%) |
| **SOCIODEMOGRAPHICS** | | |
| Age (years) | 44.2±8.2 | 44.6±6.1 |
| Gender | | |
| Male | 308 (93.3%) | 329 (99.1%) |
| Female | 22 (6.7%) | 3 (0.9%) |
| Marital status | | |
| Single/Divorced/Separated/Widowed | 143 (43.5%) | 154 (46.4%) |
| Married | 187 (56.5%) | 178 (53.6%) |
| Education | | |
| No school/Primary/Secondary school | 126 (38.0%) | 165 (49.7%) |
| High school (Grade 10–12) | 128 (38.9%) | 120 (36.1%) |
| Vocational training/College/University | 76 (23.1%) | 47 (14.2%) |
| Occupation | | |
| Unemployed/Homemaker | 12 (3.6%) | 24 (7.2%) |
| Salaried/paid jobs | 297 (90.0%) | 289 (87.0%) |
| Other (farmers, retired/students) | 21 (6.4%) | 19 (5.7%) |
| Household income in the past 12 months | | |
| More than 300 million VND | 47 (14.2%) | 26 (7.8%) |
| Less than 300 million VND | 283 (85.8%) | 306 (92.2%) |
| Living arrangement | | |
| Live alone | 33 (10.0%) | 26 (7.8%) |
| Live with spouse/children/grandchildren | 234 (70.9%) | 229 (69.0%) |
| Living with others | 63 (19.1%) | 77 (23.2%) |
| **RISK BEHAVIOR** | | |
| Illicit drug use | | |
| Never | 106 (32.1%) | 45 (13.6%) |
| Not in the past 3 months | 184 (55.8%) | 218 (65.6%) |
| In the past 3 months | 40 (12.1%) | 69 (28.8%) |
| Hazardous drinking (AUDIT-C) | | |
| No | 142 (43.0%) | 138 (41.6%) |
| Yes | 188 (57.0%) | 194 (58.4%) |
| **HEALTH & HIV TREATMENT** | | |
| Depressive symptoms (CES-D 8) | | |
| No | 205 (62.1%) | 213 (64.2%) |
| Yes | 125 (37.9%) | 119 (35.8%) |
| Self-rated health | | |
| Excellent/Very good/Good | 113 (34.2%) | 87 (26.2%) |
| Fair/Poor | 217 (65.8%) | 245 (73.8%) |
| Duration of HIV diagnosis (years) | 11.1±6.3 | 13.6±6.2 |
| Duration of ARV treatment uptake (years) | 9.3±5.2 | 10.5±5.0 |
| ARV treatment adherence level of ≥95% | | |
| No | 24 (7.3%) | 29 (8.7%) |
| Yes | 306 (92.7%) | 303 (91.3%) |

*(Continued)*

**Table 1.** (Continued)

| | Type of tobacco product use | |
|---|---|---|
| | **Cigarette-only smokers** | **Dual smokers** |
| **TOBACCO SMOKING AND CESSATION** | | |
| Smoke-free home rule | | |
| Allowed everywhere inside the home | 172 (52.1%) | 213 (64.2%) |
| Not allowed anywhere/ Allowed in some places/at sometimes | 158 (47.9%) | 119 (35.8%) |
| Provider advised for tobacco use treatment at the last visit | | |
| Not advised | 124 (37.6%) | 129 (38.9%) |
| Advised | 206 (62.4%) | 203 (61.1%) |
| Perceived risks of smoking cigarettes compared to waterpipe | | |
| Equally harmful | 211 (63.9%) | 189 (56.9%) |
| Less harmful | 22 (6.7%) | 5 (1.5%) |
| More harmful | 97 (29.4%) | 138 (41.6%) |
| Risk Perception (Range: 0–12) | 8.3±2.6 | 8.5±2.5 |
| Self-efficacy scores (Range: 0–24) | 10.6±5.1 | 9.8±5.4 |
| Fagerstrom Nicotine Dependence Category | | |
| High/Very High | 108 (32.7%) | 191 (57.5%) |
| Very low/Low/Medium | 222 (67.3%) | 141 (42.5%) |
| Intention to quit cigarette smoking | | |
| Not planning to quit | 37 (11.2%) | 43 (13.0%) |
| Planning/trying to quit | 293 (88.8%) | 289 (87.0%) |
| Intention to quit waterpipe smoking | | |
| Not planning to quit | NA | 48 (14.5%) |
| Planning/trying to quit | NA | 284 (85.5%) |
| Quit attempts (cigarettes) | | |
| Never tried to quit | 138 (41.8%) | 207 (62.3%) |
| Ever tried to quit | 192 (58.2%) | 125 (37.7%) |
| Quit attempts (waterpipes) | | |
| Never tried to quit | NA | 204 (61.4%) |
| Ever tried to quit | NA | 128 (38.6%) |
| **SOCIAL NORMS, NETWORK, SUPPORT & STIGMA** | | |
| Norms against waterpipe smoking | | |
| Disagree | 12 (3.6%) | 10 (3.0%) |
| Agree | 318 (96.4%) | 322 (97.0%) |
| Norms against cigarette smoking | | |
| Disagree | 7 (2.1%) | 12 (3.6%) |
| Agree | 323 (97.9%) | 320 (96.4%) |
| Having smokers in social network | | |
| No | 2 (0.6%) | 2 (0.6%) |
| Yes | 328 (99.4%) | 330 (99.4%) |
| Number of smokers of the five closest persons | | |
| Less than (<) 3 | 108 (32.7%) | 80 (24.1%) |
| More than (≥) 3 | 222 (67.3%) | 252 (75.9%) |
| People around object to your smoking | | |
| No | 37 (11.2%) | 34 (10.2%) |
| Yes | 292 (88.8%) | 298 (89.8%) |

*(Continued)*

**Table 1.** (Continued)

| | Type of tobacco product use | |
|---|---|---|
| | Cigarette-only smokers | Dual smokers |
| Social Support (MSPSS) (Range: 1–4) | 3.1±0.4 | 3.0±0.4 |
| Perceived internal smoking stigma scores (Range: 1–4) | 2.5±0.7 | 2.5±0.6 |

ARV: Antiretroviral therapy, AUDIT-C: Alcohol Disorders Identification test, CES-D 8: Center for Epidemiological Studies Depression Scale, HIV: Human immunodeficiency virus, MSPSS: Multidimensional Scale of Perceived Social Support.

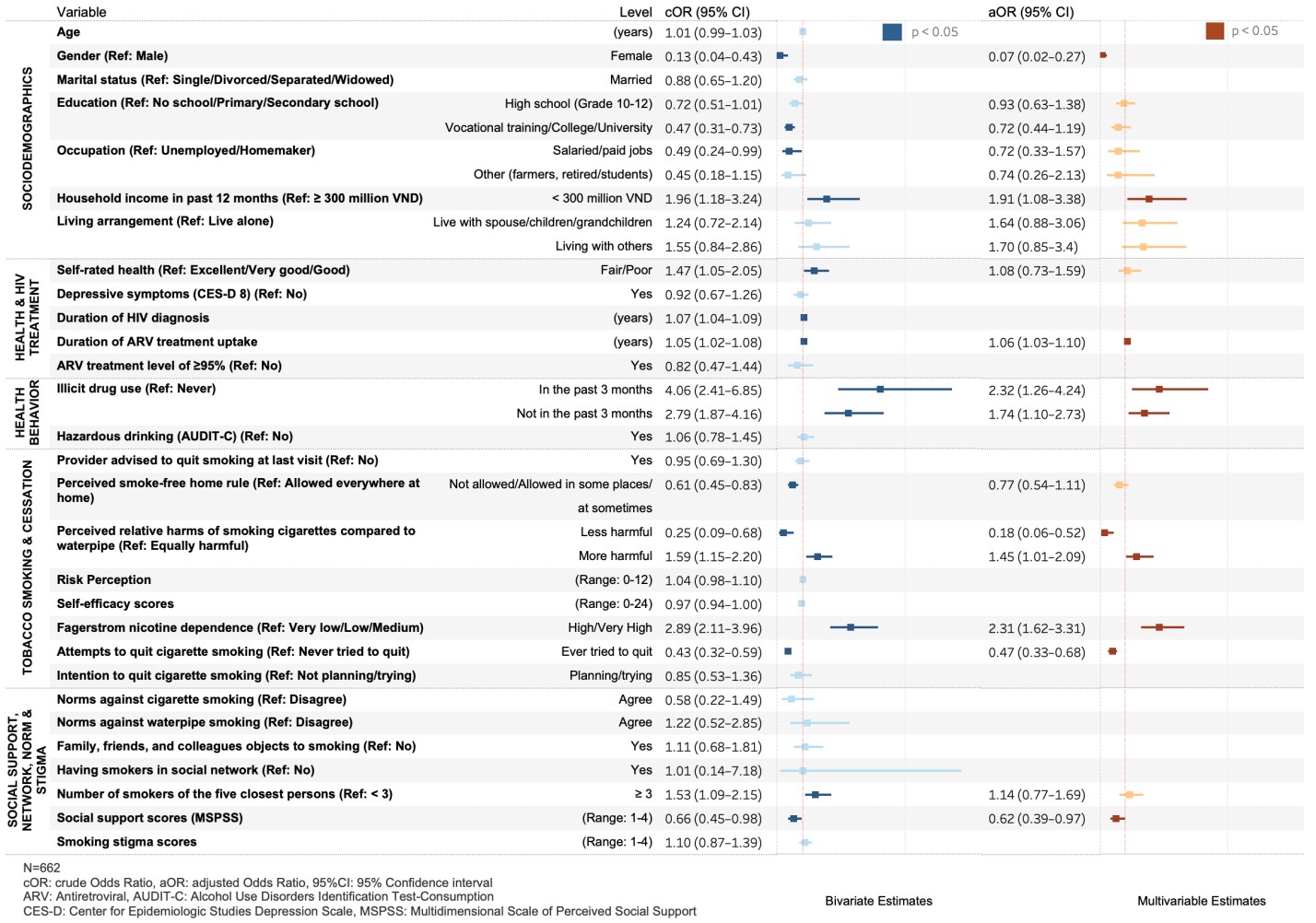

Fig 1. **Bivariate and multivariable analysis of factors associated with dual smoking.**

**Multivariable analysis.** In the multivariable model, as shown in Fig 1, several factors associated with dual smoking remained. Female cigarette smokers were less likely to also smoke waterpipes compared to males (aOR: 0.07, 95% CI: 0.02–0.27). Individuals earning less than 300 million VND had a higher likelihood of dual smoking (aOR: 1.91, 95%CI: 1.08–3.38).

Smokers who had ever used drugs but not in the last 3 months had a 74% higher likelihood of being dual smokers, compared to non-users (aOR: 1.74, 95% CI: 1.10–2.73). Among individuals who had used drugs in the past three months, the odds of dual smoking were significantly higher (aOR: 2.32, 95% CI: 1.26–4.24) compared to those who had never used drugs. There was a positive association between the duration of ARV treatment uptake and dual smoking, where a longer period was linked to a higher likelihood of being dual smokers (OR: 1.06, 95%CI: 1.03–1.10).

Smokers perceiving cigarette smoking as less harmful than waterpipe smoking (aOR: 0.18, 95% CI: 0.06–0.52) and individuals who ever made any quit attempts (aOR: 0.47, 95% CI: 0.33–0.68) were less likely to be dual smokers. Those reported high or very high nicotine dependence (aOR: 2.31, 95%CI: 1.62–3.31) were more likely to be dual smokers. Social support was associated with lower odds of dual smoking (aOR: 0.62, 95% CI: 0.39–0.97).

## Discussion

There was a high proportion of dual smoking among patients enrolled in the Vquit study. This provided an opportunity to examine, in more depth, factors associated with dual compared to cigarette-only use [25]. We found several significant differences between dual and cigarette-only smokers that highlight the complex interplay of factors influencing tobacco use patterns within this population.

Smokers living with HIV in our study, regardless of smoking type, were predominantly men, reflecting both the higher prevalence of smoking among men in the general population and deeply rooted cultural norms linking smoking with masculinity and communal social interactions [40]. Dual users had lower income compared with cigarette-only users. This is consistent with prior research that found an association between unemployment and poly-tobacco use among PLWH [15]. Our qualitative study in the same population revealed that waterpipe smoking was perceived as more affordable and socially acceptable and used as a substitute for cigarettes due to its lower cost [17]. This pattern aligns with broader evidence showing that waterpipe use is common among low-income Vietnamese smokers [41]. These findings highlight the need for tobacco control policies and interventions targeting economically disadvantaged groups.

Dual use was also associated with higher rates of drug use compared to cigarette-only smokers. Research consistently points out the interplay between polysubstance use, addictive behaviors, and the sociocultural context of waterpipe smoking [42–44]. In PLWH, the added burden of stigma, mental health challenges, and coping mechanisms may further exacerbate the likelihood of drug use among dual smokers. Addressing underlying behavioral factors such as addiction, risk-taking behaviors, and psychosocial stressors is critical, particularly for PLWH, who are already vulnerable to polysubstance use. Integrating substance use treatment into HIV care and combining these efforts with tailored tobacco cessation programs can more effectively address the co-existence of drug and tobacco use, ultimately improving health outcomes in this at-risk population. In Vietnam, although some integration exists, such as methadone maintenance treatment within HIV services, most programs still operate in silos, limiting coordinated care. Integration is most feasible at the primary care level and within HIV outpatient clinics, where routine contact with patients provides an opportunity to embed screening, brief intervention, and referral services. Moving toward integrated models will require stronger cross-program collaboration, policy alignment, and provider training to address multiple risk behaviors more effectively.

Nicotine dependence was strongly associated with dual smoking among PLWH in our study, reflecting the compounded addictive effects of using both cigarettes and waterpipes. Dual smokers demonstrated higher nicotine dependence compared to cigarette-only smokers, consistent with prior studies linking multiple tobacco product use to increased addiction severity [45–47]. Integrating tailored cessation programs within HIV care settings is essential to address this heightened dependence effectively. Additionally, public health education campaigns should emphasise the compounded health risks of dual smoking among PLWH, promoting early interventions to mitigate dependence and improve disease management outcomes.

Our findings indicated that dual smokers living with HIV were less likely to have previously attempted to quit cigarette smoking compared to cigarette-only smokers. This may reflect unique barriers to cessation within dual smokers, including

higher nicotine dependence and distinct social motivations associated with waterpipe smoking [44,47]. Since dual smokers maintain tobacco consumption through waterpipes as an alternative nicotine source, they may experience reduced urgency or motivation to quit cigarettes. Additionally, misconceptions prevalent in Vietnam that waterpipe smoking is safer or less harmful than cigarettes could lower the perceived urgency to quit cigarette smoking [11]. Our study supports this, demonstrating that dual smokers were more likely than cigarette-only smokers to perceive cigarettes as more harmful. Compounded by higher nicotine dependence resulting from dual tobacco use and psychosocial stressors unique to PLWH, such as stigma and mental health challenges, these factors collectively create significant barriers to sustained cigarette cessation attempts. Public health efforts targeting PLWH must address these misconceptions through culturally tailored campaigns that highlight the unique risks of dual smoking.

Traditional cessation programs often focus on exclusive cigarette use, potentially overlooking the dual challenges posed by waterpipe smoking. The findings suggest that PLWH who are dual users may face additional challenges to achieving abstinence. These factors underscore the need for comprehensive, integrated cessation interventions tailored to PLWH, especially in low-resource settings like Vietnam. Such interventions should include substance use management, stress and stigma reduction, and correcting misconceptions about waterpipe safety. Additionally, the absence of waterpipe-specific regulations in Vietnam, including the lack of taxes on waterpipe tobacco and health warnings, may contribute to the higher adoption of dual tobacco use in this population [48,49]. Effective control would require targeted measures, such as specific regulations on waterpipe tobacco, stronger enforcement of smoking bans for this tobacco product, public health campaigns to educate about the risks of waterpipe smoking, and a revised taxation system to reduce its appeal.

Our study addressed an often-overlooked aspect of tobacco use patterns among PLWH, contributed to the limited research on dual smoking in this group and provided actionable evidence to inform tailored cessation interventions and integrate tobacco control into HIV care. Nevertheless, several limitations should be acknowledged. The cross-sectional design limits the ability to establish causation between dual smoking and associated factors. The reliance on self-reported data introduces potential biases, such as recall and social desirability bias, which may affect the accuracy of reported smoking behaviors. The focus on PLWH receiving HIV care and enrolled in a smoking cessation program in Hanoi, Vietnam, may limit the generalisability of findings to other regions or populations, for example, those not receiving care at OPCs or those not participating in the VQUIT study. As waterpipe-only smokers were not recruited, given the study's focus on cigarette smokers, we could not identify or quantify the unique characteristics or risks associated specifically with exclusive waterpipe smoking. The proportion of female smokers in our sample was low. This reflects the real gender distribution of smoking in the study population, which aligns with national prevalence data on female smokers but limits the ability to draw firm conclusions about gender-specific patterns in dual smoking [6,50]. Additionally, the lack of longitudinal data prevents understanding changes in smoking behaviors over time, particularly regarding HIV progression and treatment and tobacco smoking cessation.

## Conclusions

This study highlighted the sociodemographic, behavioral, and social factors contributing to dual smoking among PLWH in Vietnam. These include low income, higher nicotine dependence, longer duration of living with HIV and misconceptions about waterpipe safety, emphasizing the compounded challenges faced by this population. The findings underscored the need for integrated cessation interventions within HIV care that address unique barriers such as substance use and tobacco-related misconceptions. Future research should focus on the long-term impacts of dual smoking and evaluate targeted cessation strategies for this vulnerable group.

## Supporting information

**S1 File. Inclusivity in global research questionnaire.**
(DOCX)

## Author contributions

**Conceptualization:** Thanh Ha-Lan Hoang, Gloria Guevara Alvarez, Louise Adermark, Nawi Ng, Donna Shelley.

**Data curation:** Trang Nguyen.

**Funding acquisition:** Nam Nguyen, Donna Shelley.

**Investigation:** Thanh Ha-Lan Hoang, Gloria Guevara Alvarez, Louise Adermark, Nawi Ng, Trang Nguyen, Nam Nguyen, Donna Shelley.

**Methodology:** Thanh Ha-Lan Hoang, Gloria Guevara Alvarez, Nawi Ng, Nam Nguyen, Donna Shelley.

**Project administration:** Trang Nguyen.

**Resources:** Trang Nguyen, Nam Nguyen.

**Supervision:** Gloria Guevara Alvarez, Louise Adermark, Nawi Ng, Donna Shelley.

**Validation:** Gloria Guevara Alvarez, Louise Adermark, Nawi Ng, Trang Nguyen, Nam Nguyen, Donna Shelley.

**Visualization:** Thanh Ha-Lan Hoang.

**Writing – original draft:** Thanh Ha-Lan Hoang.

**Writing – review & editing:** Thanh Ha-Lan Hoang, Gloria Guevara Alvarez, Louise Adermark, Nawi Ng, Trang Nguyen, Nam Nguyen, Donna Shelley.

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
