## [Decision Letter · Decision Letter 0]

26 Aug 2025

Dear Dr. Hoang,

Thank you for submitting your manuscript to PLOS ONE. After careful consideration, we feel that it has merit but does not fully meet PLOS ONE’s publication criteria as it currently stands. Therefore, we invite you to submit a revised version of the manuscript that addresses the points raised during the review process.

We look forward to receiving your revised manuscript.

Kind regards,

Billy Morara Tsima, MD MSc

Academic Editor

PLOS ONE

Journal Requirements:

[This study was supported by the National Cancer Institute under Award Number R01CA24481. TH received institutional funding to conduct the research at Sahlgrenska Academy.].

3. In the online submission form, you indicated that [The datasets are available from the corresponding author upon reasonable request.].

4. Please amend the manuscript submission data (via Edit Submission) to include author Thanh Ha-Lan Hoang.

5. Please amend your authorship list in your manuscript file to include author Thanh Hoang.

7. Please include a caption for Figure 1.

8. Please upload a copy of Figure 2, to which you refer in your text on page 12. If the figure is no longer to be included as part of the submission please remove all reference to it within the text.

Reviewers' comments:

Reviewer's Responses to Questions

**Comments to the Author**

1. Is the manuscript technically sound, and do the data support the conclusions?

Reviewer #1: Yes

Reviewer #2: Yes

2. Has the statistical analysis been performed appropriately and rigorously?

Reviewer #1: Yes

Reviewer #2: Yes

3. Have the authors made all data underlying the findings in their manuscript fully available?

Reviewer #1: Yes

Reviewer #2: Yes

4. Is the manuscript presented in an intelligible fashion and written in standard English?

Reviewer #1: Yes

Reviewer #2: Yes

Reviewer #1: The work demonstrates a rigorous research investigation with a defined objective, sound methods, and noteworthy findings. The study examines a pertinent public health concern and provides significant insights into dual tobacco use among PLWH in a particular low-resource environment. The manuscript is coherently organised and comprehensible.

Reviewer #2: Overall, a very well written and pertinent initiative. As most of my queries were addressed in the limitation section, I have only a few questions and some edits.

1-Were questionnaires used to assess

-Smoking risk perceptions

-Intention to quit smoking

-Internalized smoking stigma

Validated in any way prior to use?

2-Regarding recruitment, please specify if participants received a compensation and if so, what form?

Edits

In Data source section: Line 96 please define acronym NRT

In Measurement section: Line 129 ARV, Line 132 ART

In the Result section: Delete line 223, there is no Fig 2.

I commend the authors for the nicely described measurement section and Figure 1 visual.

**Do you want your identity to be public for this peer review?** For information about this choice, including consent withdrawal, please see our Privacy Policy

Reviewer #1: No

Reviewer #2: No

---

## [Author Response · Author response to Decision Letter 1]

15 Sep 2025

Responses to eviewer’s comments:

1. Were questionnaires used to assess

-Smoking risk perceptions

-Intention to quit smoking

-Internalized smoking stigma

Validated in any way prior to use?

In this revision, I cited in the manuscript the literature used to support the measurement of these variables.

Risk perception - Kaufman et al Measuring smoking cessation risk perception. NTR 2019 doi:10.1093/ntr/ntz213; https://apps.who.int/iris/bitstream/handle/10665/112836/9789241506946_eng.pdf

Smoking stigma questions were based on the Internalized Stigma of Smoking Inventory - Brown‐Johnson, C. G., Cataldo, J. K., Orozco, N., Lisha, N. E., Hickman III, N. J., & Prochaska, J. J. (2015). Validity and reliability of the internalized stigma of smoking inventory: An exploration of shame, isolation, and discrimination in smokers with mental health diagnoses. The American journal on addictions, 24(5), 410-418.

Readiness to quit - the specific citation we reference for the questions is a study that tested the model in the context of smoking -- DiClemente CC, Prochaska JO, Fairhurst SK, Velicer WF, Velasquez MM, Rossi JS. The process of smoking cessation: An analysis of precontemplation, contemplation, and preparation stages of change. Journal of Consulting.

2. Regarding recruitment, please specify if participants received a compensation and if so, what form?

The survey took approximately 45 minutes, and after its completion, the participant received an incentive of 2.5 USD. I added this in the Methods.

3. Edits

In Data source section: Line 96 please define acronym NRT

In Measurement section: Line 129 ARV, Line 132 ART

In the Result section: Delete line 223, there is no Fig 2.

---

## [Editor Report · Decision Letter 1]

21 Oct 2025

Exploring factors associated with dual tobacco smoking among people living with HIV receiving care at HIV outpatient clinics in Hanoi, Vietnam

PONE-D-25-25187R1

Dear Dr. Hoang,

We’re pleased to inform you that your manuscript has been judged scientifically suitable for publication and will be formally accepted for publication once it meets all outstanding technical requirements.

Kind regards,

Billy Morara Tsima, MD MSc

Academic Editor

PLOS ONE
---

## [Editor Report · Acceptance letter]

PONE-D-25-25187R1

PLOS ONE

Dear Dr. Hoang,

I'm pleased to inform you that your manuscript has been deemed suitable for publication in PLOS ONE. Congratulations! Your manuscript is now being handed over to our production team.

Kind regards,

on behalf of

Dr. Billy Morara Tsima

Academic Editor

PLOS ONE